# Effect of the Degree of Hydrolysis on Nutritional, Functional, and Morphological Characteristics of Protein Hydrolysate Produced from Bighead Carp (*Hypophthalmichthys nobilis*) Using Ficin Enzyme

**DOI:** 10.3390/foods11091320

**Published:** 2022-04-30

**Authors:** Kamal Alahmad, Wenshui Xia, Qixing Jiang, Yanshun Xu

**Affiliations:** 1State Key Laboratory of Food Science and Technology, School of Food Science and Technology, Collaborative Innovation Center of Food Safety and Quality Control in Jiangsu Province, Jiangnan University, Wuxi 214122, China; kamalalani85@yahoo.com (K.A.); qixingj@163.com (Q.J.); xuys@jiangnan.edu.cn (Y.X.); 2Department of Food Science and Technology, Faculty of Agriculture, University of Alfurat, Deir Ezzor, Syria

**Keywords:** enzymatic hydrolysis, ficin enzyme, bighead carp, molecular weight, DSC thermal property, functional properties

## Abstract

The production of fish protein hydrolysates from bighead carp *(Hypophthalmichthys nobilis)* using ficin enzymes was achieved in optimal conditions of 3% enzyme/substrate ratio, 40 °C temperature, and pH 6. Three different hydrolysis times, 1, 3, and 6 h, were investigated, and their degree of hydrolysis (DH) values were 13.36%, 17.09%, and 20.15%, respectively. The hydrolysate yield values increased with DH increase, and the highest yield was obtained at DH 20.15%. The crude protein content increased from 80.58% to 85.27%, and amino acid compositions increased from 78.33% to 83.07%. The peptides formed during hydrolysis indicated low molecular weight that might improve functional characteristics of fish protein hydrolysates, including protein solubility, which ranged from 84.88% to 95.48% for all hydrolysates. The thermal degradation of hydrolysates occurred from 160 to 168 °C with intensive endothermic peaks. Results revealed that oil holding capacity was higher at DH 13.36%; water holding capacity was higher when DH increased. Hence, fish protein hydrolysates (FPH) from bighead carp have improved functional properties, and can be utilized as supplements and excellent protein sources in various uses in food applications.

## 1. Introduction

The bighead carp is endogenous from North to South China rivers, including Han Chiang, Min, and Yangtze Rivers. This species spread from Asia to other places, including North America, and it is major global aquaculture with a global production of around 3150 kilotons in 2018 [1]. In Indiana, fish carp were reported early in rivers such as Wabash, Tippecanoe, and Ohio Rivers. However, the body of a bighead carp is laterally compressed with a head that is close to one-third of its body size. 

Bighead carp fish are silver on the sides and the back, with gray to creamy on their bellies. Bighead carp is a very valuable and popular ingredient in China, and the quality of bighead carp fish is a primary concern for producers and consumers [2]. They filter rivers and lakes through their comblike rakers, only consuming small organisms to penetrate and pierce their filter-feeding system. They feed on various organisms such as water plants, insect larvae, algae, fish eggs, and insects. Fish carp are very tolerant of challenging conditions; different carp fish have a strong sense of taste, smell, and hearing, and tend to gather in small groups [3]. 

Most markets prefer bighead carp to weigh from 8 to 14 pounds, but larger or smaller fish can cost much less. Bighead carp competes with other species in the marketplace, although bighead substitutes affect the market price. Moreover, fish protein is highly digestible, and a rich source of peptides and essential amino acids that are limited in other meat proteins. In addition, fish proteins have remarkable biological values due to amino acids in sufficient proportions [4]. Carp fish is a rich source of protein because of its high nutritional value and high protein content. Bighead carp contains enough amino acids, fatty acids in abundance, and vitamins, including riboflavin (B2), pyridoxine (B6), cyanocobalamin (B12), and D3 [5]. Protein hydrolysis is a process to obtain peptide fractionation containing a mixture of amino acids [6]. Different techniques might be applied to obtain protein hydrolysates, such as ultrasound and traditional chemical techniques or other organic solvents. Hydrolysis using acids is a conventional method to obtain protein hydrolysates; a strong alkaline or acid separates peptide bonds at high temperature and pressure.

However, enzymatic protein hydrolysis has great application potential in food industries compared to other hydrolysis techniques. This process can give potential nutritional and functional products for various uses in the food industry. Protein enzymatic hydrolysis is a process to enhance various properties of original proteins. In addition, this method effectively provides and promotes small active peptides that can be applied as antioxidants, anti-cancer with regulate the immune system and enhance the body functions such as blood pressure [4]. Different protein fractions of peptides have recently been applied to food, cosmetics, and other areas [7]. The enzymatic proteolysis of proteins obtained from various sources with solubilization and functional properties was deeply investigated and reported by various researchers over the past 30 years [8]. Many enzymes were applied for protein hydrolysis, such as trypsin, chymotrypsin, flavourzyme, alcalase, ficin, papain, neutrase, pancreatic, pronsae, validase, orientase, thermolysin, and bromelain [6].

Previous enzymes are outstanding examples of protease enzymes used in biocatalysts and synthesis. Protease enzymes exist in all microorganisms, animals, and plants. They are applied to many industrial fields in food biotechnology, including for sensory properties such as taste and flavor, the production of bioactive peptides, minimizing allergies, and enhancing digestibility [9]. Ficin enzymes obtained from *Ficus carica* trees is a protease that catalyzes the ends of cysteine, which is similar to papain bromelain and cathepsin. These enzymes belong to the family of cysteine proteases with the exact same mechanism of catalysis and same cleaving peptide bonds [10]. Ficin can produce hydrolysates from different protein sources and be applied in food industries, such as for enhancing the texture and tenderness of fish products. Ficin is also used as a milk clotter, a rennet starter, and a processing path for fermented cereals [11]. 

Enzymatic hydrolysis, compared with other technical hydrolysis, provides the best conditions for hydrolysis to obtain high-quality products of fish protein hydrolysate FPH. Enzymatic hydrolysis could be a suitable and appropriate technique to improve the characteristics of FPH products and maintain their nutritional values for food applications [7]. The hydrolytic process is related to different factors that affect the final hydrolysate product, including the concentration of enzymes, substrate, reaction time, and reaction temperature. Moreover, the optimal conditions of enzymatic factors can render the process more effective in producing the desired FPH [12]. 

FPH could improve functions used in the food industry compared with untreated fish protein. These enhanced functions are foaming capacity, water, and oil holding capacity, emulsification property, and antioxidative, antimicrobial, and anticancer activities [13]. The substrate with the type of protease enzymes used to obtain FPH significantly impacts hydrolysate products’ functional and nutritional properties. Protein hydrolysates are widely used in different areas as healthy food, and could be used as an anticancer drug or to prevent different gastrointestinal diseases and protect the human body from them [7]. In addition, FPH can be used as a protein supplement for building muscles and as food additive applied to different baked goods or meat products [6]. The production of bioactive peptides is an important field in using protease enzymes such as ficin, papain, or bromelain. Protease enzymes were used to produce FPH from Indian fish, and hydrolysate properties were evaluated as anti-breast cancer agents and for other medical treatments [14]. 

Here, we examine for the first time the influence of ficin enzymatic hydrolysis on the degree of hydrolysis and production of FPH from bighead carp under optimal conditions (substrate/enzyme ratio, pH, and temperature), which had not previously been investigated, in addition to evaluating the nutritional morphological and functional properties of hydrolysate products based on the variance of the degree of hydrolysis. 

## 2. Materials and Methods

### 2.1. Raw Fish Muscle Preparation

Fish bighead carp samples were acquired through the aquatic science and technology industry (Yangtze River, Wuxi, China). Samples were transported in an iced Styrofoam container directly to the aquatic product and food processing research laboratory at the School of Food Science. Specimens were cleaned, and fins, bones, and heads were removed. Then, they were stored frozen at −20 °C until further use. 

### 2.2. Enzyme and Reagents 

Ficin enzyme (enzyme activity 610 MCU/mg, pH 6.0, temperature 40 °C) was purchased from Wuxi Decheng Lebang Biotechnology Co., Ltd. (99 Jinxi Road, Binhu, Wuxi, China). Then, the enzyme was kept at 4 to 5 °C until the start of the experiments. Other chemicals and solutions, including alkaline and acids, were supplied by Sinopharm Reagents Company (Wuxi, China). The rest of the reagents and compounds were of analytical grade.

### 2.3. Protein Hydrolysate Preparation 

Fish protein hydrolysates of bighead carp were produced using ficin enzyme following the procedure explained by Noman [7] with minor modification in temperature and pH conditions (Table 1). Minced fish muscles were mixed with 25 mM buffer (sodium phosphate) to maintain a steady pH throughout hydrolysis. Steps of the hydrolytic process are detailed in Figure 1. Enzymatic hydrolysis was performed in a jacketed reaction vessel (250 mL) containing a stirrer (IKA-MAG Hs 4, MS 25) and linked with a constant circulated water bath (Blue pard Technical. Co., Shanghai, China) to control the temperature, which remained steady during hydrolysis. Hydrolytic samples were heated to 70 °C for 20 min in the water bath to inactivate enzymatic activity. The mixture was then cooled to room temperature and centrifuged at 10,000 rpm for 20 min at 4 to 5 °C. The supernatant was then collected and freeze-dried at a −60 °C vacuum (Scientz-10ND, Scientz Biotechnol Co., Ltd., Ningbo, China); the hydrolysate powder was then stored at −18 °C for further experiments.

### 2.4. Determination of Hydrolysis Degree 

The degree of hydrolysis is the percentage of the free amino group cleaved from hydrolysate samples, and it was determined through formal titration following the procedure in [7] with slight modification. A detected amount of protein hydrolysate samples (1.5 to 1.7 g) was added to 45 to 50 mL purified water. The mixture was calibrated to pH 7.0 by adding 0.1 N NaOH. Then, 10 mL of formaldehyde 37% (*v*/*v*) was added and kept for 8 min at room temperature. Using 0.1 N of NaOH standard solution, titration was carried out to the endpoint at pH 8.5. The amount of the free amino group was then calculated using the consumed volume of NaOH. The Kjeldahl technique was used to determine total nitrogen (TN) [15]. The degree of hydrolysis and free amino groups were calculated as follows:Free amino groups (%) = (V × C × 14.007/M × 1000) × 100(1)
Degree of hydrolysis (DH) = (% Free amino groups/% total nitrogen) × 100(2)
where V = volume consumed of NaOH (mL); C = the concentration of NaOH (0.1 M); M = the quantity of hydrolysate in the sample (g).

### 2.5. Chemical Composition Analysis 

Raw muscles and protein hydrolysate samples produced under enzymatic hydrolysis were individually investigated. AOAC standard methods 985.2912, 960.39, 928.08, and 923.03 were used to determine the contents of moisture, fat, protein, and ash, respectively [16]. In brief, 1 to 2 g of the samples was weighed in the crucible and burned in a muffle furnace at 500 to 550 °C for 6 h until constant weight and reddish ash were observed. The sample (2 to 3 g) was extracted in a solvent (petroleum ether) for 3 to 4 h in Soxhlet lipid extraction equipment (Lipid Auto determiner, Xian Jian Instruments Co., Shanghai, China). The crude protein was evaluated using the Kjeldahl analyzer (DK-3400/FOSS, Hilleroed, Denmark) and multiplying total nitrogen obtained by 6.25 (standard factor). Moisture content was determined by evaporating samples for 3 h at 100 to 105 °C until they had reached a consistent weight.

### 2.6. Yield 

FPH yield was calculated by the weight of FPH obtained as a percentage of fish muscles (substrate) and was used in enzymatic hydrolysis according to Romadhoni [17] with the following equation:Yield (%) = [weight of FPH (g)/weight of Raw materials (g)] × 100(3)

### 2.7. Color Measurement

A Hunter Lab digital colorimeter (UltraScan PRO, D65, Hunter Lab, Reston, VA, USA) was used to scale the color of the protein hydrolysate products. Processed samples were at the port of the colorimeter. Values of color evaluation L*, a*, and b* were measured; a* represents redness to green; b* represents yellowness to blue; L* represents lightness; and ΔE expressed the total color difference. 

### 2.8. Amino Acid Determination

The standard method of AOAC was applied to analyze the amino acid profile of the hydrolysate samples [18]. In brief, 100 mg of protein hydrolysate samples was digested by using 8 mL of 6 M HCl at 110 °C under a nitrogen atmosphere in an air oven for 20 to 23 h. In room temperature, processed samples were then cooled, and 4.8 mL of 10 M NaOH was added. Samples were washed using purified water up to 25 mL. Samples were filtered using Whatman filter paper and centrifuged for 15 min at 10,000. Amino acids were analyzed and separated using HPLC (model 1100, Agilent Technologies, Santa Clara, CA, USA); the reverse phase mode was 180 mm × 4.6 (Agilent) with Zorbax 80 A and C18 column at 40 °C, and detection was carried out at 338 nm with a constant flow rate of 1 mL/min. Mobile Phase A was 7.35 mM/L sodium acetate/triethylamine/tetrahydrofuran (500:0.12:2.5, *v*/*v*/*v*); to adjust pH to 7.2, acetic acid was used. Mobile Phase B was 7.35 mM/L sodium acetate/methanol/acetonitrile (1:2:2, *v*/*v*/*v*) with pH 7.2. Analysis of amino acids was expressed as a gram of amino acid per 100 g of the protein sample.

### 2.9. Molecular-Weight Distribution Profile 

The molecular-weight distribution of bighead carp fish muscles and the protein hydrolysate obtained by enzyme hydrolysis at 1,3 and 6 h were analyzed according to the guidance of Liu, Lin, Lu, and Cai [19] with a minor modification. High-performance liquid chromatography using a Waters-1525 (Milford, MA, USA) system with column TSKgel 2000 SWXL (300 mm × 7.8 mm; Tosoh, Tokyo, Japan) was equilibrated with mobile-phase acetonitrile/water/trifluoroacetic acid (TCA) 45/55/0.1 (*v*/*v*). Samples before injection were dissolved in a mobile-phase buffer and centrifuged for 20 min at 8500 rpm before filtration using a 0.45 µm filtrate membrane. Lastly, samples were eluted at a 0.5 mL/min flow rate and monitored at 220 nm. The temperature of the column was 30 °C. Cytochrome C (12,384 Da), bacitracin (1422 Da), Gly-Gly-Try-Arg (451 Da), and Gly-Gly-Gly (189 Da) were utilized as standards to prepare the calibration curve of molecular mass analysis.

### 2.10. Scanning Electron Microscopy 

The morphological properties of the hydrolysate products were implemented using scanning electron microscope analysis (Hitach-High-Tech’s SU1510 Minato Ku, Tokyo, Japan). Treated samples were coated before being loaded into the SEM machine, and images were then observed under accelerating voltage of 1.0 Kv with a secondary electron image. The image was scanned using a 10.20 mm Ricoh Camera with 600× magnification.

### 2.11. Thermal Characterization Analysis

FPH was exposed to a differential scanning calorimeter (DSC) (DSC-Q200-V24.8 Build 120, TA instrument, New Castle, DE, USA) according to the method reported by Alahmad, Xia, Jiang, and Xu [20] with some modification. Hydrolysate samples (25 ± 3 mg) were compacted, sealed in aluminum pans, and gradually heated in a DSC at temperatures ranging from −25 to 175 °C at a flow rate of 10 °C/min. Samples were then maintained for 5 min at 175 °C. After that, samples were cooled to −25 °C at a flow rate of 10 °C/min. As a control sample, an empty aluminum pan was used.

### 2.12. Functional Properties Analysis of FPH

#### 2.12.1. Solubility 

The solubility of the protein hydrolysate obtained by enzymatic hydrolysis was evaluated on the basis of the recorded procedure by Jain and Anal [21] with slight changes. In total, 20 mL purified water and 200 mg of hydrolysate were mixed, and the pH solution was adjusted from 2 to 10 by utilizing 0.1 M NaOH or 0.1 M HCl. Solutions were incubated at 30 °C with stirring at 150 rpm for 30 min, and then centrifuged at 8000 g for 20 min. Protein content in both supernatant and hydrolysate samples was determined using the Kjeldahl process [15], and the following equation calculated solubility:Solubility % = (protein content in the supernatant/total protein content in the sample) × 100(4)

#### 2.12.2. Emulsifying Properties

The emulsification properties of FPH under various degrees of hydrolysis, including emulsifying stability index (ESI) and emulsifying activity index (EAI), were determined according to Noman, Xia, and others [22] with minor modification. Different concentrations of protein hydrolysate solutions (0.1%, 0.5% and 1% protein, 6 mL) were mixed with 10 mL of soybean oil and homogenized (ULTRA-TURRAX, T 18 Digital, IKA, Germany) at 22,000 rpm for 60 s. After emulsion formation, 50 µL was collected from the bottom of the samples and diluted with 5 mL of 0.1% sodium dodecyl sulfate solution. Solution absorbance was detected at 500 nm utilizing a UV 1000 spectrophotometer after 0 and 10 min. The following equations were used to calculate EAI and ESI:EAI (m^2^/g) = 2 × 2.303 × A_500_ × DF/IΦC(5)
ESI (min) = A_500_ × 10/∆A(6)
where A_500_ = absorption value at 500 nm, Φ = oil volume fraction (0.25), I = path length of the cuvette (m), DF = dilution factor, C = protein concentration in aqueous phase (g/m^3^), ∆A = A_0_ − A_10._

#### 2.12.3. Water Holding Capacity

Water holding capacity (WHC) of three different protein hydrolysates was determined according to a protocol described by [12] with some changes. We dissolved 0.5 g of hydrolysate samples in distilled water (10 mL) and dispersed it using a vortex for 50 s. The dispersion was maintained for 6 h at room temperature (25 °C) and was centrifuged at 4000 rpm for 30 min. After that, the supernatant was filtered using Whatman filter paper no. 1, and the recovered volume was determined. Lastly, the difference between the initial volume of distilled water and the supernatant volume was calculated. WHC values were recorded as mL of water absorbed per gram of hydrolysate product.

#### 2.12.4. Oil Holding Capacity

The oil holding capacity value of different protein hydrolysates samples was measured according to the method reported by [12] with minor modifications. In total, 10 mL of soybean oil was mixed with the hydrolysate samples (0.5 g) and vortexed for 60 s in a centrifuge tube. The sample was then centrifuged at 4000 rpm for 30 min and accurately measured. The absorption of hydrolysate samples was measured on the basis of weight variance, and the oil holding capacity (OHC) value was given in grams of oil absorbed per gram of protein hydrolysate.

### 2.13. Statistical Analysis

Samples were analyzed in triplicate (*n* = 3), and data are provided as mean value ± standard deviation (±SD). Results were subjected to one-way analysis of variance (ANOVA) to show the significant differences. Duncan’s test was performed to analyze multiple ranges between means utilizing SPSS v. 19 (SPSS, Chicago, IL, USA); significance level was *p* ≤ 0.05.

## 3. Results and Discussion

### 3.1. Production of Protein Hydrolysates

In the current work, the impact of ficin enzymatic hydrolysis conditions on the DH is demonstrated in Figure 2. The optimal pH was 6, and the optimal temperature was 40 °C during the enzymatic hydrolysis process. The effect of E/S ratio on DH was experimentally investigated at various levels according to protein content in the substrate (fish muscles), namely, 1%, 2%, 3%, 4%, and 5%. At 1% (*w*/*w*) ficin enzyme concentration, the DH was 13.56%, which could have been due to inadequate catalytic sites that are important to improve the degree of hydrolytic value. Enzyme concentration increased to 3%, and the DH value was 16.96%, which showed significant difference compared to the low concentration of the ficin enzyme.

The higher concentration of the ficin enzyme at 4% and 5% presented no significant variance in DH values, likely due to the enzyme aggregation, which inhibited active sites to the catalytic protein in the substrate during hydrolysis [7]. The optimal E/S ratio was 3%, which was similar to the optimal E/S ratio reported by Noman [7] using a papain enzyme with substrate Chinese sturgeon fish. On the other hand, under optimal conditions of the ficin enzyme (pH = 6, *t* = 40 °C) and E/S 3%, which was experimentally investigated, the DH was determined at three different times (1, 3, and 6 h) to be 13.36%, 17.09%, and 20.15%, respectively. The DH increased during hydrolysis from 1 to 3 h, and continued to 6 h. Presented data were approximately higher than those reported by [23], who used papain enzyme, similar to those determined by Noman and others using 2.4 L Alcalase enzyme protease [22].

### 3.2. Proximate Chemical Composition

Proximate analysis of bighead carp fresh muscles and the protein hydrolysates is presented in Table 2. Moisture content in the fresh muscles was 77.31%, which was similar to results reported by Noman, Xu, AL-Bukhaiti, Abed, Ali, Ramadhan, and Xia [7]. Protein, fat, and ash in fresh muscles were 15.78%, 1.68%, and 1.34, respectively. Fat content was lower than that in [7], but higher than the fat value reported by [12]. Furthermore, the proximate composition of fish protein hydrolysate is given in Table 2. During enzymatic hydrolysis by the ficin enzyme, the content of protein hydrolysate increased, namely, 85.27% at DH 20.15% (6 h), followed by 81.93% at DH 17.09% (3 h); the lowest value of 80.58% was recorded at DH 13.36% (1 h). However, difference in DH led to various ranges of protein content in the hydrolysates; thus, increasing DH showed increases in protein percentage in the samples, which may be related to the elimination of insoluble and indigested fragments during enzymatic hydrolysis [24].

However, lipid content in the enzymatic hydrolysates under 13.36%, 17.09%, and 20.15% of DH was 4.13%, 4.59%, and 2.84%, respectively. Table 2 shows a significant difference in fat content between hydrolysates obtained at 1 and 3 h compared to the hydrolysate obtained at 6 h. Noman and others [22] reported fat values ranging from 7.9% to 12% in protein hydrolysate obtained from Chinese sturgeon fish. Data in Table 2 show the opposite relation between protein percentage and fat content in the obtained hydrolysates, which was reported by [24].

In addition, the increase in fat content in the hydrolysates may regard lipid particles that existed in the supernatant after the sample centrifuge. Ash content of hydrolysates at DH 13.63% was significantly different from ash content at DH 17.09% and 20.15%; these values were similar to those reported by [22], and lower than the ash content presented by [23].

### 3.3. Yield

Yield data in this work are presented in Table 2. Under optimal conditions of enzymatic hydrolysis, the highest yield value of the hydrolysate was recorded at DH 20.15%, with significant difference compared to those obtained at DH 17.09% and 13.63%. Noman and others reported that the yield value ranged from 13.5% to 17.4% using Alcalase 2.4 L enzyme under various DH [22].

### 3.4. Color Measurements

Color assessment data (L*, a*, b*) are demonstrated in Table 2. Hydrolysate samples lean towards whiteness (L* = 89.23, 88.82, 89.55) and a little yellowness (b* = 10.96, 12.41, 12.68) for enzymatic hydrolysis samples obtained at DH 13.36%, 17.09%, and 20.15%, respectively. According to the results, no significant differences were noticed between processed samples at different DHs. However, in the current study, the L* value was higher than that reported by Noman, Qixing, Xu, Ali, Al-Bukhaiti, Abed, and Xia [22], but the b* value was a little bit lower in the same study. Color values in protein hydrolysates could be attributed to the composition of fish muscles with enzymatic hydrolysis conditions to produce the hydrolysate powder [25].

### 3.5. Amino Acid Analysis

The amino acid concentrations of fish protein hydrolysates obtained by using the ficin enzyme under three different hydrolysis times are shown in Table 3. Amino acids are important biomolecules that regulate metabolic pathways and improve the synthesis of important biological substances [4]. The total amino acids of the degree of hydrolysis of 13.36%, 17.09%, and 20.15% were 78.33, 80.36, and 83.07 g/100 g protein, respectively; these values were significantly different (*p* < 0.05). By increasing the time of enzymatic hydrolysis, the DH and content of the amino acids both increased, which was attributed to small peptides forming during the hydrolysis process, which led to the concentration of total amino acids increasing. Major amino acids found in the treated samples were aspartic, glutamic, lysine, and leucine. Total amino acid values in the current work were higher than those reported by Noman, Qixing, Xu, Ali, Al-Bukhaiti, Abed, and Xia [22], but lower than the values of the amino acids studied by Ovissipour et al. [26]. Additionally, sufficiently high amounts of essential amino acids could play crucial roles in protein synthesis; however, the evaluation of protein quality is related to amino acid compounds [20]. Therefore, the enzymatic hydrolysis process might enhance the content of essential amino acids in protein hydrolysates compared to raw materials.

### 3.6. Molecular-Weight Distribution

Analysis of the molecular-weight profile of the unhydrolyzed sample and FPH produced from Chinese bighead carp at various DHs using ficin enzyme under optimal conditions is shown in Figure 3. Data showed that protein hydrolysates obtained at different DHs possessed small peptides compared to the unhydrolyzed sample, which could be related to the enzymatic hydrolysis process and its influence on high-molecular-mass peptides. Peptides of various molecular weights were found in the protein hydrolysates obtained at 13.36%, 17.09%, and 20.15%. The molecular-weight distribution of <1000 Da showed 82.11%, 86.71%, and 95.49% of FPH obtained at DHs of 13.36%, 17.09%, and 20.15%, respectively. Increasing hydrolysis time led to an increase in DH, and more peptides with low molecular weight were then released during the enzymatic hydrolysis process.

However, the proportion of molecular mass with less than 1000 Da in FPH of Chinese sturgeon using the Alcalase enzyme obtained at various DH ranged from 66% to 80% [22]. Our results were higher than those reported by Rivero-Pino et al. [27], while the percentage of molecular weight <1000 Da in protein hydrolysate derived from silver carp by using Alcalase was around 60% [28]. In 2016, Stizyte, Rommi, and others reported that using protease enzymes to obtain protein hydrolysates from salmon could decrease the number of substantial molecular weight from >5000 Da to small-molecular-mass peptides (150 to 500 Da) [29]. Moreover, molecular weight <1000 Da values were higher in our study than those by Zhang et al. [30], who recorded a higher molecular weight of >5000 Da using different commercial proteases.

### 3.7. Scanning Electron Microscopy Analysis

The study of untreated and FPH samples obtained from Chinese bighead carp using ficin enzyme utilizing SEM is shown in Figure 4. Results in the SEM images showed that, after enzymatic hydrolysis, significant peptides broke down into small particles, which explains the reduction in particle size in the samples of protein hydrolysates at DH 13.36%, 17.09%, and 20.15% (Figure 4A–C, respectively) compared to the untreated sample (Figure 4D). On the other hand, SEM micrographs in the treated samples showed that some particles were rectangular with acute angles, which could be attributed to the effects of the enzymatic hydrolysis process and evaporation during the freeze-drying technique. However, our assertions are approximately correlated to [31], which reported that huge peptides during enzymatic treatment degraded into tiny particles, and the size of the molecular samples was reduced.

### 3.8. Differential Scanning Calorimeter Analysis

The thermal properties of protein hydrolysates obtained at three different DHs are demonstrated in Figure 5. Three treated samples exhibited a temperature range of −25 to 175 °C with a constant flow rate of 10 °C/min. Protein hydrolysates at DH 13.36%, 17.09%, and 20.15% were exposed to slight thermal decomposition at 86.08, 91.76, and 80.57 °C, respectively. However, after the first decomposition, all samples showed a stable curve before being exposed to different thermal degradation levels at different temperatures of FPH samples. During the DSC process, sharp thermal degradation occurred at 160 and 165 °C of the protein hydrolysate obtained at DH 13.36%. However, at DH 20.15%, sharp thermal decomposition with deep endothermic peaks was observed at 162 and 168 °C. The endothermic response of the treated samples may indicate the enzymatic hydrolysis conditions and peptides obtained by the hydrolysis process. Ortiz and Anon (2001) reported two slight endothermic peaks at 79.9 and 95.5 °C in soybean protein hydrolysates [32]. The presented work revealed that the overshadowing of thermal peaks could be attributed to overlapping endothermic curves. The hydrolytic process had an evident influence on the transition of enthalpy and temperature of hydrolysate samples. Liu and his colleagues reported endothermic peaks ranging from 56 to 105 °C in fish protein hydrolysates at different degrees of hydrolysis [33].

### 3.9. Functional Properties of FPH

#### 3.9.1. Protein Hydrolysate Solubility

The solubility of hydrolysate samples at various DHs and in the pH range from 2 to 10 are given in Figure 6. The solubility of protein hydrolysates considerably impacts protein and peptide functions. It is essential in the food industry to use protein products in applications of emulsions and foams to enhance their functional properties [34]. Presented data showed the high solubility of the hydrolysate samples, ranging from 85% to 95% at various pH levels, and the top value of solubility was 95.5% at pH 6 and DH 20.15% with significant variance (*p* < 0.05). No significant difference appeared in solubility values at DH 17.09% and 20.15%. Enzymatic hydrolysis degraded huge peptides into small-molecular-weight peptides, resulting in more solubility in the protein products [35]. pH ranged from 4.5 to 5.5 at isoelectric protein point. The protein was less soluble because of less interaction between it and water. Solubility increases due to the high interaction between water and protein [36], supporting our findings of high protein solubility at pH 2, 6, and 8. Previous studies reported similar results. Noman and others [22] determined the solubility of hydrolysates using the Alcalase enzyme from Chinese sturgeon fish. Li and colleagues [37] also reported the solubility values of protein hydrolysates produced from grass carp using the papain enzyme. Additionally, variation in the solubility values of FPH at different DHs could be attributed to the peptides’ size and the ions of the peptides group forming during the hydrolysis process [38].

#### 3.9.2. Water and Oil Holding Capacity

WHC and OHC results of FPH produced from Chinese bighead carp using a ficin enzyme under optimal conditions are demonstrated in Table 2. Previous studies revealed that protein hydrolysates have great WHC values that enhance the flavor of different meat meals [39]. Presented results show a significant difference (*p* < 0.05) at different DHs. The top value of WHC was at DH 20.15%, followed by DH at 17.09% and 13.36%. The current results of WHC were similar to those reported by [12], but higher than values reported by Noman [22]. The increase in DH during enzymatic hydrolysis led to an increase in polar groups COOH and NH2 in the hydrolysate samples, which influenced the values of water absorption capacity [39].

OHC values were 3.85, 3.72, and 2.94 g/g FPH at DHs of 13.36%, 17.09%, and 20.15%, respectively. Approximately lower values of OHC compared to our values were determined by Wasswa and others when obtaining hydrolysates from grass carp skin using Alcalase [12]. The reduction in OHC values and the increase in DH could be related to the molecular weight of peptides that formed during hydrolysis, as smaller molecular weight could absorb less oil. Therefore, the lowest value of OHC was recorded at DH of 20.15%. OHC is a fundamental functional property that affects the emulsifying properties and sensory of food products.

#### 3.9.3. Emulsifying Properties

Table 4 shows the emulsifying activity index (EAI) and emulsion stability index (ESI) of FPH prepared from Chinese bighead carp using enzymatic hydrolysis with different concentrations (0.1%, 0.5%, and 1%) at three different DHs (13.36%, 17.09%, and 20.15%). Emulsion is the absorption of protein particles during homogenization to the surface of composed oil droplets to build a shielding membrane to protect oil droplets from merging. EAI values at hydrolysate concentration of 0.1% were 92.75, 76.82, and 72.04 m^2^/g at DH 13.36%, 17.09% and 20.15%, respectively. In addition, ESI values were significantly higher at a concentration of 0.1% compared to those at 0.5% and 1% protein concentrations at various DHs.

The increase in protein concentration and DH showed a decrease in EAI and ESI records, which was an inverse correlation. These findings are supported by those of Nalinanon and others [38], who showed a similar influence on the relationship between DH at different protein concentrations, and EAI and ESI values from ornate threadfin bream using the pepsin enzyme. Noman and his colleagues observed the same impact regarding emulsifications and DH at various hydrolysate concentrations [22]. Protein hydrolysates with small or hydrophobic peptides might improve emulsification stability [40]. Obtained peptides from enzymatic hydrolysis with low molecular mass might not be amphiphilic enough to show significant emulsion properties. Emulsion characteristics could be attributed to protein hydrolysate concentrations, and the formed hydrolyzed peptides’ size and molecular weight.

## 4. Conclusions

Fish protein hydrolysates were produced from bighead carp *(Hypophthalmichthys nobilis)* using a ficin enzyme as a biocatalyst at optimal conditions (pH: 6, temperature: 40 °C, and enzyme/substrate: 3%). Results showed that nutritional and functional properties were significantly influenced by the degree of hydrolysis. The percentage of total essential amino acids to total amino acids ranged from 51.97% to 52.85%. Moreover, functional properties, including the emulsion activity, emulsion stability, solubility, and water and oil holding capacity of the hydrolysates showed good enhancement in terms of applications in food industries.

High values of WHC and OHC were obtained at DH 20.15% and 13.36%, respectively. The current study provided adequate analysis of the morphological and thermal properties of hydrolysate samples. Additionally, the produced hydrolysates had remarkable nutritional properties with vast potential to be used as protein supplements in food processing. Peptide fractionations, and their antioxidant and antimicrobial properties should be studied in future analysis.

## Figures and Tables

**Figure 1 foods-11-01320-f001:**
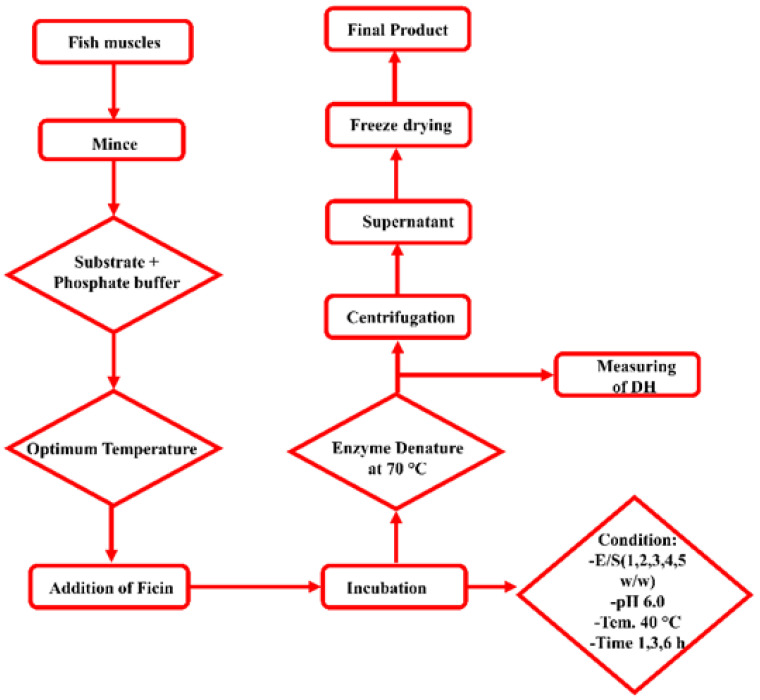
Scheme of hydrolysate production from bighead carp using ficin enzyme.

**Figure 2 foods-11-01320-f002:**
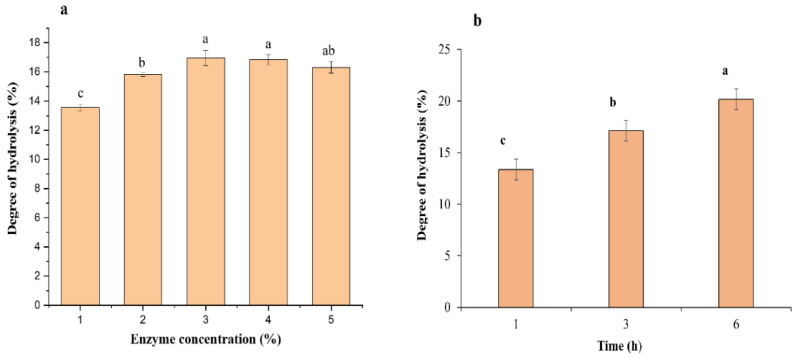
(**a**); Effect of enzyme/substrate ratio, (**b**); hydrolysate production under three different hydrolytic times. Mean ± SD (*n* = 3). Different letters indicate significantly different values (*p* ≤ 0.05). All results represent the mean of triplicate determinations.

**Figure 3 foods-11-01320-f003:**
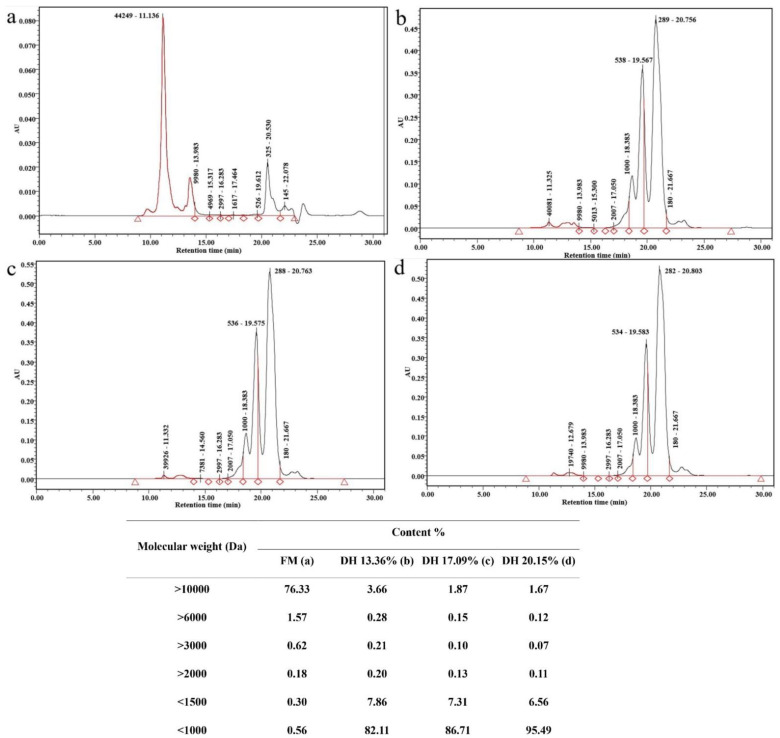
Molecular-weight distribution of raw and protein hydrolysates obtained from bighead carp using ficin enzyme under different times of hydrolysis: (**a**) fresh sample; (**b**) DH 13.36% (1 h); (**c**) DH 17.09% (3 h); and (**d**) DH 20.15% (6 h).

**Figure 4 foods-11-01320-f004:**
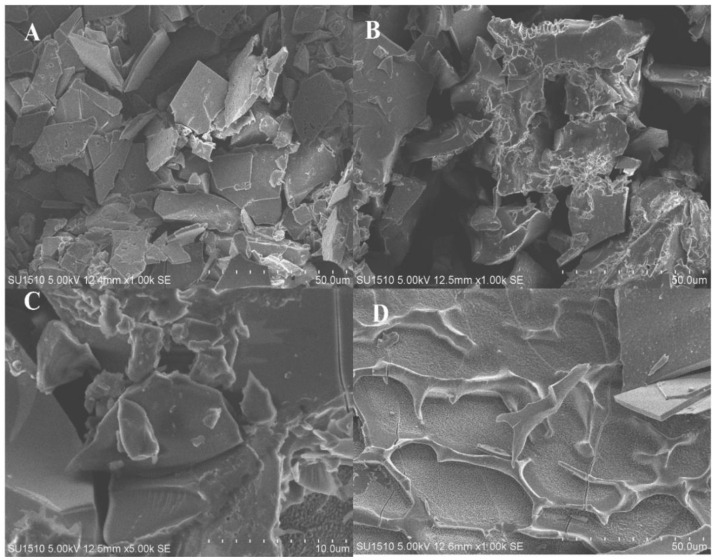
Scanning electron microscopy (SEM) of fish protein hydrolysates during different hydrolysis times and with fresh sample. (**A**) DH 13.36%, (**B**) DH 17.09%, (**C**) DH 20.15%, and (**D**) fresh fish sample.

**Figure 5 foods-11-01320-f005:**
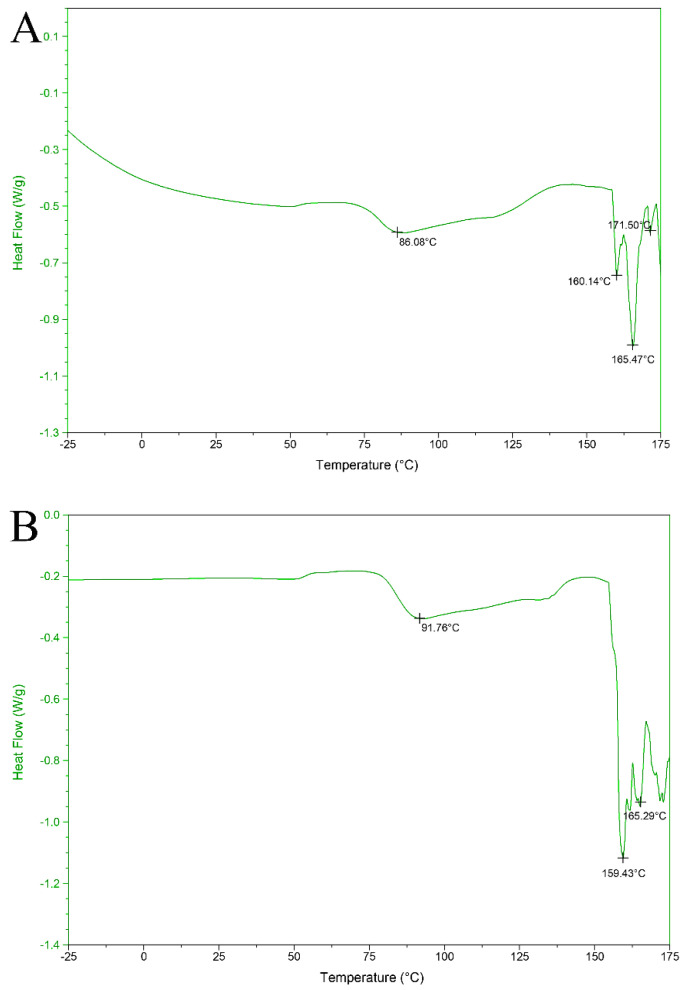
Differential scanning calorimetry of bighead carp protein hydrolysates at various degrees of hydrolysis. (**A**) DH 13.36%, (**B**) DH 17.09%, and (**C**) DH 20.15%.

**Figure 6 foods-11-01320-f006:**
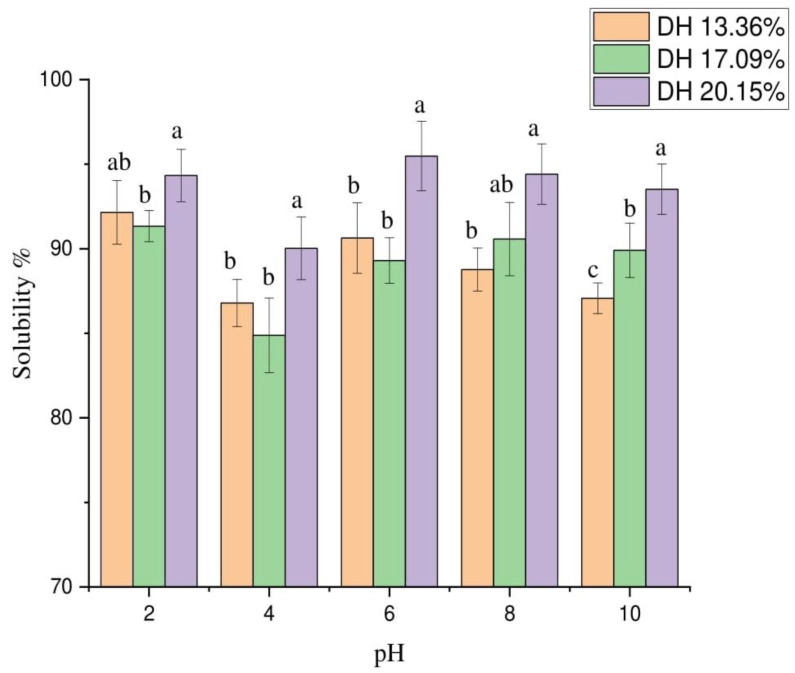
Protein solubility of hydrolysates at different degrees of hydrolysis at pH 2–10. Values represent mean ± SD (*n* = 3), and various letters indicate that values are significantly different (*p* ≤ 0.05).

**Table 1 foods-11-01320-t001:** Enzymatic hydrolysis conditions for obtaining protein hydrolysates from bighead carp *(Hypophthalmichthys nobilis)* using ficin enzyme.

Hydrolysis Conditions	Ficin Enzyme
Enzyme/substrate ratio (%)	1, 2, 3, 4, 5
Reaction pH	6
Temperature (°C)	40
Inactivation temperature of hydrolysis (°C)	70
Inactivation time (min)	20
Hydrolysis time (h)	1, 3, 6

**Table 2 foods-11-01320-t002:** Proximate composition, color measurement, yield, and WHC and OHC of bighead carp fresh muscles and protein hydrolysates (FPH) prepared using ficin enzyme (*n* = 3, mean ± SD).

Parameters		Hydrolysates		
	Fresh Sample	DH 13.36% (1 h)	DH 17.09% (3 h)	DH 20.15% (6 h)
Moisture	77.31 ± 0.93	4.26 ± 0.09 ^a^	4.21 ± 0.06 ^a^	4.28 ± 0.06 ^a^
Protein	15.78 ± 0.25	80.58 ± 0.65 ^c^	81.93 ± 1.15 ^b^	85.27 ± 0.73 ^a^
Fat	1.68 ± 0.07	4.13 ± 0.43 ^a^	4.59 ± 0.28 ^a^	2.84 ± 0.12 ^b^
Ash	1.34 ± 0.03	6.69 ± 0.18 ^a^	6.04 ± 0.11 ^b^	5.97 ± 0.16 ^b^
Color determination				
L*	-	89.23 ± 0.08 ^a^	88.82 ± 0.10 ^b^	86.55 ± 0.09 ^c^
a*	-	0.17 ± 0.01 ^b^	0.23 ± 0.05 ^ab^	0.31 ± 0.06 ^a^
b*	-	10.96 ± 0.56 ^b^	12.29 ± 0.10 ^a^	12.68 ± 0.03 ^a^
∆E	-	19.85 ± 0.47 ^a^	18.71 ± 0.25 ^b^	18.92 ± 0.16 ^b^
Hydrolysate yield	-	11.25 ± 0.16 ^c^	14.67 ± 0.20 ^b^	15.92 ± 0.31 ^a^
WHC (g/g FPH)	-	2.18 ± 0.15 ^c^	2.79 ± 0.21 ^b^	3.93 ± 0.19 ^a^
OHC (g/g FPH)	-	3.85 ± 0.10 ^a^	3.72 ± 0.13 ^a^	2.94 ± 0.17 ^b^

Different letters indicate significantly different values (*p* ≤ 0.05). All values represent the mean of triplicate determinations, mean ±SD (*n* = 3). L*, a*, b*, ΔE values are standards of color measurements. L* refers to lightness ranging from 0 to 100, a* refers to green to red, and b* refers to blue to yellow. WHC, water holding capacity (g water/g hydrolysate). OHC, oil holding capacity (g oil/g hydrolysate).

**Table 3 foods-11-01320-t003:** Amino acid content of bighead carp protein hydrolysates (g/100 g protein) derived using ficin enzyme under optimal conditions.

Amino Acids	Hydrolysates		
Essential Amino Acids (EAAs)	DH 13.36% (1 h)	DH 17.09% (3 h)	DH 20.15% (6 h)
Histidine	2.63 ± 0.02 ^b^	2.65 ± 0.02 ^b^	2.78 ± 0.05 ^a^
Threonine	3.13 ± 0.06 ^c^	3.24 ± 0.05 ^b^	3.36 ± 0.11 ^a^
Valine	4.23 ± 0.06 ^b^	4.32 ± 0.02 ^b^	4.43 ± 0.04 ^a^
Methionine	2.39 ± 0.16 ^b^	2.65 ± 0.02 ^a^	2.77 ± 0.10 ^a^
Phenylalanine	3.05 ± 0.02 ^c^	3.22 ± 0.01 ^b^	3.29 ± 0.02 ^a^
Isoleucine	3.80 ± 0.05 ^c^	3.86 ± 0.01 ^b^	4.01 ± 0.03 ^a^
Leucine	6.42 ± 0.05 ^c^	6.51 ± 0.02 ^b^	6.74 ± 0.02 ^a^
Lysine	8.12 ± 0.08 ^b^	8.01 ± 0.06 ^b^	8.49 ± 0.06 ^a^
Arginine	4.99 ± 0.01 ^b^	4.97 ± 0.01 ^b^	5.17 ± 0.07 ^a^
Tyrosine	2.03 ± 0.22 ^c^	2.34 ± 0.05 ^b^	2.62 ± 0.03 ^a^
**Nonessential amino acids (NAAs)**			
Cystenie	0.23 ± 0.03 ^b^	0.28 ± 0.01 ^b^	0.36 ± 0.02 ^a^
Aspartic acid	9.72 ± 0.07 ^c^	10.00 ± 0.03 ^b^	10.45 ± 0.19 ^a^
Glutamic acid	13.79 ± 0.25 ^c^	14.02 ± 0.05 ^b^	14.46 ± 0.10 ^a^
Serine	2.26 ± 0.04 ^c^	2.59 ± 0.05 ^b^	2.70 ± 0.04 ^a^
Glycine	4.33 ± 0.07 ^a^	4.21 ± 0.04 ^b^	4.09 ± 0.01 ^c^
Proline	2.48 ± 0.02 ^a^	2.54 ± 0.04 ^a^	2.28 ± 0.02 ^b^
Alanine	4.70 ± 0.12 ^b^	4.91 ± 0.08 ^a^	5.01 ± 0.07 ^a^
TEAA	40.81 ± 0.45 ^c^	41.79 ± 0.29 ^b^	43.69 ± 0.36 ^a^
TNAA	37.52 ± 0.17 ^c^	38.57 ± 0.11 ^b^	39.38 ± 0.82 ^a^
TAA	78.33 ± 1.12 ^c^	80.36 ± 0.73 ^b^	83.07 ± 0.68 ^a^
TEAA/TAA%	51.97 ± 0.28	52.04 ± 0.41	52.85 ± 0.15

Different letters indicate that values are significantly different (*p* ≤ 0.05); values represent the mean of triplicate determinations, mean ± SD (*n* = 3). TAA, total amino acids; TEAA, total essential amino acids; TNAA, total nonessential amino acids; TEAA/TAA %, total essential amino acids to total amino acids.

**Table 4 foods-11-01320-t004:** Emulsifying characteristics of protein hydrolysate obtained from bighead carp using ficin enzyme with different hydrolysate concentrations at various DHs (mean ± SD, *n* = 3).

DH %	Concentration %	EAI (m^2^/g)	ESI (min)
13.36	0.1	92.75 ± 1.39 ^a^	51.39 ± 1.28 ^a^
	0.5	27.36 ± 0.68 ^b^	29.94 ± 1.09 ^b^
	1	10.68 ± 0.51 ^c^	23.46 ± 0.75 ^c^
17.09	0.1	76.82 ± 2.07 ^a^	45.87 ± 0.91 ^a^
	0.5	14.98 ± 0.82 ^b^	20.16 ± 0.41 ^b^
	1	8.22 ± 0.45 ^c^	18.40 ± 0.34 ^c^
20.15	0.1	72.04 ± 1.69 ^a^	36.03 ± 0.86 ^a^
	0.5	16.11 ± 0.97 ^b^	19.78 ± 0.73 ^b^
	1	8.17 ± 0.65 ^c^	11.83 ± 0.24 ^c^

Different letters in the same column of each concentration and degree of hydrolysis indicate significantly different values (*p* ≤ 0.05).

## Data Availability

The data presented in this study are available in this article.

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
