# Peer review of "Effect of the Degree of Hydrolysis on Nutritional, Functional, and Morphological Characteristics of Protein Hydrolysate Produced from Bighead Carp (Hypophthalmichthys nobilis) Using Ficin Enzyme"

_foods, 2022, doi:10.3390/foods11091320_

Round 1

Reviewer 1 Report

Alahmad and co-worker produced fish protein hydrolysates by using ficin enzyme at pH 6, 40°C.  The enzyme/substrate ratio was investigated in the range of 1 to 5% (w/w), and 3% was found the optimal one. Keeping the 3% E/S ratio constant, the Authors investigated different incubation times (1, 3 and 6 h). The obtained hydrolysates were characterized in term of degree of hydrolysis, chemical composition, yield, colour, amino acid composition, molecular weight distribution, morphological properties, thermal properties, solubility, emulsifying properties, WHC, and OHC.

The manuscript was prepared with very little care and attention. For example, information on the different incubation times is missing in Table 1, while information on the E/S ratio is missing in Figure 1. Paragraph 3.4 should be the first in the Results section, as it contains results on the E/S ratio that are preliminary to the results on the different incubation time. On page 14 there is an image, containing 4 chromatograms without legend, never mentioned in the text.

Apart from this general lack of attention, which requires a thorough and careful review of the manuscript by the authors, there is in my opinion a major methodological flaw, which must be addressed before proceeding with any other comment on the results.

In enzymatic treatments, the enzyme/substrate ratio must be defined unambiguously in term of activity units. The Authors tested different percentage of enzyme (w/w). According to Materials and Methods, line 116, Ficin used by the Authors has a specific activity ranging from 400 to 1000 MCU/mg. This information means that the product they used has a Ficin content potentially very different for each batch of product. In other word, it is impossible to define the amount of activity units used by the Authors. To allow the reproducibility of the experiments, and to reliably define the proposed treatments, the Authors must provide the specific activity (MCU/mg) of the lot used. Alternatively, they must include in their study an enzymatic assay suitable for characterizing the specific activity of the enzyme preparation. In the absence of this information, in my opinion, the whole study has relative significance.

Author Response

Point–by-point response

Authors' Reply to the Review Report (Reviewer 1)

Comments

1- The manuscript was prepared with very little care and attention. For example, information on the different incubation times is missing in Table 1, while information on the E/S ratio is missing in Figure 1. 

Response: Thank you for your careful check of the manuscript, the manuscript has been carefully revised, and missing data has been added according to your suggestions.

2- Paragraph 3.4 should be the first in the Results section, as it contains results on the E/S ratio that are preliminary to the results on the different incubation time.

Response: Thanks for your comment; we do appreciate your careful revision; we did change and modify paragraph 3.4 and became paragraph 3.1 in the results and discussion part.

3- On page (14) there is an image, containing 4 chromatograms without legend, never mentioned in the text.

Response: Thank you very much for your valued comment; we apologize for this error while filling the manuscript in the journal template. The 4 chromatograms are parts of figure 3 ( the figure has been corrected on page 11).

4- Apart from this general lack of attention, which requires a thorough and careful review of the manuscript by the authors, there is in my opinion a major methodological flaw, which must be addressed before proceeding with any other comment on the results.

Response: Thank you for your careful review, the manuscript has been carefully revised, and the shortcomings in all parts have been addressed as recommended.

5- In enzymatic treatments, the enzyme/substrate ratio must be defined unambiguously in term of activity units. The Authors tested different percentage of enzyme (w/w). According to Materials and Methods, line 116, Ficin used by the Authors has a specific activity ranging from 400 to 1000 MCU/mg. This information means that the product they used has a Ficin content potentially very different for each batch of product. In other word, it is impossible to define the amount of activity units used by the Authors. To allow the reproducibility of the experiments, and to reliably define the proposed treatments, the Authors must provide the specific activity (MCU/mg) of the lot used. Alternatively, they must include in their study an enzymatic assay suitable for characterizing the specific activity of the enzyme preparation. In the absence of this information, in my opinion, the whole study has relative significance.

Response: Thanks for your valuable comments, the specific activity of the enzyme batch used in the study was added as it was 610 MCU/mg (page 3, line 117).

Reviewer 2 Report

The article entitled " Effect of the degree of hydrolysis on Nutritional, functional, and morphological characteristics of protein hydrolysate produced from bighead carp (Hypophthalmichthys nobilis) using  ficin enzyme" written by Alahmad et al., are found very much significant and interesting. Alahmad et al., carried a study on bighead carp fish  for the development of protein hydrolysates using enzymes and characterizes its nutritional as well as functional properties.

However, going through the manuscript, I found that, it lacks several important points that needs to be discussed. Here are some remarks that have to be considered in further submission.

General comment:

  • At several places there is spacing issue, authors are suggested to correct it
  • Author should carefully follow the journal guidelines; referencing pattern should be thoroughly checked.
  • Authors are recommended to choose more scientific words rather than general vague term to describe any important aspects.
  • Over all, English language should be thoroughly checked, and at some places, grammar mistakes are present that need to be correct to improve the quality of the manuscript.

Specific comment:

  • Abstract- Authors are suggested to provide initial before any abbreviation throughout the manuscript including abstract.
  • Introduction: While presenting the introduction, objective of study is not very clear. Authors are suggested to rewrite the introduction with more specificity with respect to objectivity.
  • Few references provided in the manuscript is very old, kindly update with recent references. One of the recently published article ( Drugs 2020, 18, 265; doi:10.3390/md18050265) on fish and its hydrolysate can be discuss and cited in the introduction and discussion section, this will improve the overall scientific impact of the manuscript.
  • Authors are suggested to keep uniformity, while presenting the citation in text.
  • The discussion of the present research needs further explanation and further explanations and interpretations must be added in discussion section.
  • Most of the comments has been marked in the pdf file, kindly refer to pdf pages

Author Response

Point–by-point response

Author’s Reply to the Review Report (Reviewer 2)

Responses on PDF Comments

1- DH should be introduced first.

Response: Thank you very much for your great suggestion. I did re-write the meaning of (DH), page (1) line 14-15.

2- Hydrolysates' thermal degradation has occurred.

Response: Thank you for your careful check, it has been modified, page (1) line 20

3- (FPH) products from bighead carp, “should be introduced”.

Response: Thank you for your concern, the meaning of FPH has been introduced, page (1) line 23.

4- Rephrase the sentence (and the back and grey to).

Response: Thank you for your comment, it has been rephrased, page (1), line 36.

5- Bighead carp contains enough levels of amino acids (using appropriate words).

Response: Thank you for your valuable advice, it has been modified according to your advice, page (2), line 50.

6- Kindly check the word format or follow Journals guidelines.

Response: Thank you for your valuable comment, it has been modified according to journal format, page (2), line 50 to 56.

7- Hydrolysis using Acids is considered, (using small letter).

Response: Thanks for your careful check, the sentence has been modified, page (2) line 54.

8- Recently different protein fractions of peptides have been applied in food, cosmetics, or other application areas, (reference needed).

Response: Thanks for your comment, the reference has been added, page (2), line 64.

9- Enzymatic proteolysis of proteins from different fields, (recheck the sentence).

Response: Thanks for your careful revision, the sentence has been modified, page (2), line 65.

10- Many enzymes have been applied for protein hydrolysis, (add such as).

Response: Thanks for your advice, it has been added, page (2), line 67.

11- Kindly check the word format or follow Journals guidelines.

Response: Thanks for your comment, the format has been checked, page (2), line 67-69.

12- Ficin enzyme obtained from (Ficus carica), should be Italic.

Response: Thanks for your check, it has been corrected, page (2), line 74.

13- FPH (should be introduced first).

Response: Thanks for your comment, it has been introduced, page (2), line 82

14- The hydrolysis process related to different factors affected its final product, (kindly check, better to keep in present tense).

Response: Thanks for your comment, it has been modified, page (2), line 85.

15- (From different gastrointestinal diseases), (reference is required).

Response: Thanks for your comment, the reference has been added, page (3), line 96.

16- (Temperature and pH conditions Table1), (need space).

Response: Thanks for your comment, the space has been added, page (3), line 125.

17- Line 133-135, check the format of the journal.

Response: Thanks for your advice, it has been checked according to the journal format.

18- Write full abbreviation of DH, page (4), line 141.

Response: Thanks for your careful revision, the full abbreviation of DH has been added, page (4) line 141.

19- (and then10 mL), spacing issue.

Response: Thanks for your careful check, it has been corrected, page (4) line 145.

20- Why Soxhlet lipid extraction method for raw muscles and protein hydrolysate were used?? Does the AOAC no. mentioned justify this procedure for fish products? Kindly provide the details procedure for fat extraction in protein hydrolysates and raw muscle of fish. Does Soxhlet is appropriate methods for fat extract in fish. Kindly justify.

Response: Thanks for your great comment, we have been modified the AOAC no of fat extraction (960, 39) page (4), line 156. Some researches related to fish protein hydrolysis used Soxhlet for fat extract and here are some of these researches:

(Influence of enzymatic hydrolysis conditions on the degree of hydrolysis and functional properties of protein hydrolysate obtained from Chinese sturgeon (Acipenser sinensis) by using papain enzyme). Author: Anwar Noman, journal: Process Biochemistry

(Influence of Degree of Hydrolysis on Chemical Composition, Functional Properties, and Antioxidant Activities of Chinese Sturgeon (Acipenser sinensis) Hydrolysates Obtained by Using Alcalase 2.4L). Author: Anwar Noman, journal: Journal of Aquatic Food Product Technology

(Degree of hydrolysis, functional and antioxidant properties of protein hydrolysates from Grass Turtle (Chinemys reevesii) as influenced by enzymatic hydrolysis conditions).

Author: Md. Serajul Islam, journal: Food Science & Nutrition (Wiley).

21- Provide the reference of amino acids determinations.

Response: Thanks for your comment, the reference has been added, page (5), line 176-177.

22- (The detection value was 338 nm with a constant flow rate), check the sentence.

Response: Thanks for your kind check, it has been corrected, page (5), line 184.

23- (According to the guidance of Liu, Lin, Lu, and Cai), It was guided or it is reference method?

Response: Thanks a lot for your valuable comment, it was a reference method and it has been corrected "According to Liu, Lin, Lu, and Cai"

24- Fish protein hydrolysates (FPH) were exposed, (Introduction should be done at first point of writings).

Response: Thanks a lot for your valuable advice, it has been modified, page (6), line 211.

25- And then heated in DSC, (introduce first).

Response: Thanks a lot for your comment, it has been modified based on your recommendation, page (6), line 214.

26- Keep the uniformity in maintaining the space while citing and writing.

Response: Thanks a lot for your careful revision, they have been checked according to your recommendations.

27- Kindly check and follow the guidelines of Journals while citing in text.

Response: Thank you very much for your careful revision, the citations have been checked and modified.

28- Change caption as table.

Response: Thank you very much for your valued check, all of the captions have been checked.

29- Authors are suggested to discuss the results with recent articles.

Response: Thank you very much for your valued advice, some recent references (2020 and 2022) related to molecular weight distribution have been added, pages (10-11), lines (387, 392-394).

30- Authors are suggested to compare results with some other types of fish hydrolysates rather than different kind of matrix.

Response: Thank you very much for your valued advice, the referred reference has been deleted and then added anew reference related to fish protein hydrolysate, page (12), line 427-428.

31- How the authors were able to set the optimum conditions parameters as mentioned in conclusion. What was the parameter to select pH-6, temp-40C and enzyme/substrate-3%? Kindly justify.

Response: Thank you very much for your valued comment, actually, the optimum temperature 40 ℃ and optimum pH = 6 were given by the enzyme producing company and it was mentioned in the manuscript, however, about enzyme/substrate ration was optimized experimentally to get the optimum ratio and then was found to be 3%.

32- The great values of WHC and OHC, (choose the appropriate words).

Response: Thank you very much for your valued suggestion, they have been modified, page (15), line 511.

33- Authors are suggested to provide latest reference.

Response: Thank you very much for your great suggestion, some recent references have been added according to your valuable suggestion. We appreciate your hardworking in revising the manuscript. Modified references (1, 2, 7, 12 and 38).

Round 2

Reviewer 1 Report

The authors corrected the main methodological flaws. However, they still show a lack of attention in the preparation of the manuscript.

The following points should be further clarified / corrected.

Lane 115: change “-20” in “-20°C”

Lane 117: change “PH” in “pH”

Lane 120: Did the authors use “alkaloids” for their research?

Lane 125: change “25Mm” in “25 mM”

Lanes 131-132: “The hydrolysis samples were heated to 75 ℃ for 15 min in the water bath to inactivate the enzyme activity.” Table 1 shows 75°C for 20 minutes, whereas Figure 1 shows 70°C for enzyme denaturation. Please be consistent

Lane 146: change “(1.5 to 1.7) g” in “1.5 to 1.7 g” or "(1.5 to 1.7 g)".

Lane 180: “hydrochloride-HCl”, delete “hydrochloride”

Lane 182: “then added 4.8 mL of (10 M NaOH).” Why is "10 M NaOH" in parentheses?

Lane 189: “The mobile phase (B) was 7.35 mM/L sodium acetate/methanol/acetonitrile (1:2:2, v/v/v) with pH 7.2.” What does it mean “mM/L”?

Lane 223: “20 mL purified water with (200 mg) of hydrolysate was mixed,…”. Why is "200 mg" in parentheses?

Lanes 292-294: “Figure 2. Effect of enzyme/substrate ratio: (a) the hydrolysate production under three different hydrolysis times (b).”. The description of panels (a) and (b) is unclear. Please correct.

Lanes 302-303: “the composition of fish protein hydrolysate was given in table 2.”. What "composition" do the authors refer to? Please clarify.

Lanes 353-355: “By increasing the time of enzymatic hydrolysis, the DH increased, and the composition of total amino acids increased,…”. What "composition" do the authors refer to? Please clarify.

Lanes 374-375: “The peptides present in the inactive forms in protein chains are activated after hydrolysis using protease enzymes [4].”. This sentence is out of context in this paragraph. Please delete it.

Figure 3: The quality of figure 3 is in my opinion too low, the chromatograms appear to be screenshots. What is the unit of measurement of the x-axis? What do the numbers on the chromatograms mean?

Lane 431: “Liu and his colleagues reported endothermic peaks ranging from 56 °C to 1o5 °C…”. Change “1o5” in “105”.

Fugure 5: The quality of figure 5 is in my opinion too low, the thermograms appear to be screenshots.

Lane 510-511: “The hydrolysates products have sufficient quantities of essential amino acids (51.9% to 52.85%).”.  “Sufficient” has a relative meaning, which needs to be contextualized. Please provide a reference to support the claim.

Author Response

Point–by-point response

Authors' Reply to the Review Report (Reviewer 1- Round 2)

Comments

1- Lane 115: change “-20” in “-20°C”

Response: Thank you for your careful check of the manuscript, we did modify, page (3), line 115.

2- Lane 117: change “PH” in “pH”

Response: Thank you for your careful revise, we did change it, page (3), line 117.

3- Lane 120: Did the authors use “alkaloids” for their research?

Response: Thank you for the comment, actually it was alkaline (NaOH), we did modify in the manuscript, page (3), line 120.

4- Lane 125: change “25Mm” in “25 mM”.

Response: Thank you for your careful check, we did change it, page (3), line 126.

5- Lanes 131-132: “The hydrolysis samples were heated to 75 ℃ for 15 min in the water bath to inactivate the enzyme activity.” Table 1 shows 75°C for 20 minutes, whereas Figure 1 shows 70°C for enzyme denaturation. Please be consistent.

Response: Thank you for your careful revision, we did modify to make it consistent at 70°C for 20 min, page (3), line 132 and we did change on table 1 as well.

6- Lane 146: change “(1.5 to 1.7) g” in “1.5 to 1.7 g” or "(1.5 to 1.7 g)".

Response: Thank you for your comment, we did modify in the manuscript page (4), line 146.

7- Lane 180: “hydrochloride-HCl”, delete “hydrochloride”

Response: Thank you for your comment, the word “hydrochloride” has been deleted, page (5), line 180.

8- Lane 182: “then added 4.8 mL of (10 M NaOH).” Why is "10 M NaOH" in parentheses?

Response: Thank you for your careful check, the parentheses have been deleted, page (5), line 182.

9- Lane 189: “The mobile phase (B) was 7.35 mM/L sodium acetate/methanol/acetonitrile (1:2:2, v/v/v) with pH 7.2.” What does it mean “mM/L”?

Response: Thank you for your careful check, “mM/L” means (millimoles per liter) according to amino acids analysis method.

10- Lane 223: “20 mL purified water with (200 mg) of hydrolysate was mixed,” Why is "200 mg" in parentheses?

Response: Thank you for your careful revise, the parentheses have been deleted, page (6), line 224.

11- Lanes 292-294: “Figure 2. Effect of enzyme/substrate ratio: (a) the hydrolysate production under three different hydrolysis times (b).” The description of panels (a) and (b) is unclear. Please correct.

Response: Thank you for your careful check of the manuscript, the description of panels (a) and (b) have been modified and corrected, page (8), line 292-294.

12- Lanes 302-303: “the composition of fish protein hydrolysate was given in table 2.” What "composition" do the authors refer to? Please clarify.

Response: Thank you for your careful revise of the manuscript, it means the proximate composition, the sentence has been modified in line 302-303, and the title of table 2.

13- Lanes 353-355: “By increasing the time of enzymatic hydrolysis, the DH increased, and the composition of total amino acids increased,” What "composition" do the authors refer to? Please clarify.

Response: Thank you for your careful revise of the manuscript, it means the content of the amino acids, the sentence has been modified in line 353, and the title of table 3.

14- Lanes 374-375: “The peptides present in the inactive forms in protein chains are activated after hydrolysis using protease enzymes [4].” This sentence is out of context in this paragraph. Please delete it.

Response: Thank you for your careful check of the manuscript, the sentence you refereed to has been deleted, page (10).

15- Figure 3: The quality of figure 3 is in my opinion too low, the chromatograms appear to be screenshots. What is the unit of measurement of the x-axis? What do the numbers on the chromatograms mean?

Response: Thank you for your careful check of the manuscript, we tried our best to make figure 3 clear in the revised manuscript, the unit of measurement of the x-axis is (min) because this axis is the retention time, and the numbers on the chromatograms describe the molecular weight values (Da) with the values of retention time for each peptide.

16- Lane 431: “Liu and his colleagues reported endothermic peaks ranging from 56 °C to 1o5 °C…” Change “1o5” in “105”.

Response: Thank you for your careful revision of the manuscript, we did change it, page (13), line 430.

17- Figure 5: The quality of figure 5 is in my opinion too low, the thermograms appear to be screenshots.

Response: Thank you for your careful check of the manuscript, we tried our best to appear the figure 5 and make all the thermographs clear.

18- Lane 510-511: “The hydrolysates products have sufficient quantities of essential amino acids (51.9% to 52.85%).” “Sufficient” has a relative meaning, which needs to be contextualized. Please provide a reference to support the claim.

Response: Thank you for your careful check and revise, the sentence was not accurate, it has been paraphrased in page (17), line 511-512.

Reviewer 2 Report

The article entitled " Effect of the degree of hydrolysis on Nutritional, functional, and morphological characteristics of protein hydrolysate produced from bighead carp (Hypophthalmichthys nobilis) using  ficin enzyme" written by Alahmad et al., are found very much significant and interesting. Alahmad et al., carried a study on bighead carp fish  for the development of protein hydrolysates using enzymes and characterizes its nutritional as well as functional properties.

Authors have made the suggested changes, I have no further comment. 

Author Response

No comments from Reviewer 2 about my amnuscript at the minor revision.